# Factors Affecting Employees' Problem-Solving Skills in Technology-Rich Environments in Japan and Korea

**Chyul-Young Jyung, Yoowoo Lee \*, Sunyoung Park, Eunhye Cho and Romi Choi**

Department of Vocational Education and Workforce Development, Graduate School of Seoul National University, Seoul 08826, Korea; cyjyung@snu.ac.kr (C.-Y.J.); sny218@snu.ac.kr (S.P.); digh546@snu.ac.kr (E.C.); Romi@snu.ac.kr (R.C.)
\* Correspondence: youfmc@snu.ac.kr

**Abstract:** This study conducts an analysis about the impact of basic background, cultural capital, skill use, and participation in training on employees' problem-solving proficiency in Japan and Korea based on data from the Programme for the International Assessment of Adult Competencies Survey (PIAAC). This research compared four clusters (basic background, cultural capital, participation in training, skill usage) to determine the factors affecting problem-solving skills in technology-rich environments (PSTRE) in Japan and Korea. In addition, we examined whether aging moderate the relationship between skill usage and participation in training and PSTRE. The finding shows that PSTRE is associated with the basic background, cultural capital, and skill usage. Moreover, the interaction effect between the use of skill at home and age is statistically significant in Japan. Our results provide new insights for vocational psychology and work-life research in the context of employers, employees, as well as policymakers.

**Keywords:** employees' problem-solving proficiency; skill usage; aging

## 1. Introduction

The latest industrial revolution, Industry 4.0, has led to significant reforms in the economic and social environment with regard to information and communication technology (ICT), thereby transforming individuals' lives. Due to technological developments, computers and other ICT tools have been integrated into daily lives [1], and it has become necessary for individuals to be able to use these tools effectively. In the workplace, automation has substituted routine tasks through the use of digital technologies, and employees are increasingly required to perform non-routine tasks that computers cannot easily perform [2]. Jobs requiring problem-solving skills for abstract tasks are expected to increase [3,4]. Hence, these changes create the need for problem-solving skills in technology-rich environments (PSTRE), entailing "abilities to solve problems for personal, work, and civic purposes by setting up appropriate goals and plans, and accessing and making use of information through computers and computer networks" [5]. Consequently, in today's rapidly changing environment, PSTRE are among the most essential employee competencies [6].

With the growing importance of PSTRE, it has become necessary to shed light on the factors influencing such skills. Previous studies have identified socio-demographic (e.g., gender, age, immigrant status, education attainment), family-related (e.g., parents' education attainment, the number of books at home, skill usage at home), and work-related factors (e.g., PT at work, skill usage at work, job characteristics) influencing PSTRE [1,7–9]. However, research on the linkage between learning and skill development and PSTRE is scarce. Skills can be converted from accumulated cultural capital within individuals' sociocultural contexts through pedagogic action [10] or acquired directly through participation in training [11]. Learning by doing is known to play an important role in skill

development. Against this backdrop, the current research investigates the relative impact of the following factors on PSTRE: Cultural capital, participation in training, and skill usage. Additionally, since participation in training and skill usage are lifelong processes, the interaction effect between these two factors and age is analyzed.

This research explored PSTRE and its factors with regard to employees in Japan and Korea. Although Japan and Korea have geographic, societal, and cultural similarities, the level of skills differs between these two countries [12]. A cross-country comparative view may help identify the commonalities and differences among the factors enhancing problem solving in TRE in Japan and Korea. This research used data from the Programme for the International Assessment of Adult Competencies (PIAAC). The PIAAC data provide information on adults' proficiency in literacy, numeracy, and problem solving in TRE considering various background characteristics for over 40 participating countries [12]. Most related research has focused on the United States and European countries [1,7,13,14]; few studies have focused on Asian countries. This research takes a significant first step in understanding PSTRE in the context of Asia, particularly Japan and Korea, and offers critical practical implications that will be useful in improving employees' problem-solving proficiency.

### 1.1. Cultural Capital

Bourdieu [10] defines cultural capital as a mechanism of class reproduction with economic and social capital through "the form of capital". He introduced this concept to explain the elite's advantageous access to better schools and jobs, which is not accounted for by human capital (talent and skills) alone. According to Bourdieu [10], the attitudes, features, and behaviors of working-class students are incongruent with those of middle-class students. Students from privileged backgrounds encounter similar cultural experiences, which helps them advance in terms of the general culture, linguistic skills, knowledge, and other components of the educational system [15,16]. On the other hand, students from disadvantageous socioeconomic backgrounds have difficulty adjusting to school and have a higher chance of failing a class. Cultural capital can have lasting effects on the life of an individual. Gustafsson [17] reports that performance differences in PIAAC are strongly related to Programme for International Student Assessment (PISA) achievement trends in literacy and numeracy, which suggests that the effect of success in school on competence can persist in the long term.

Cultural capital exists in three forms: embodied, objectified, and institutionalized. Embodied cultural capital is the natural way of thinking, judgment, and preferences that an individual develops at home. Objectified cultural capital refers to cultural goods such as pictures, books, dictionaries, and instruments. Institutionalized cultural capital implies high educational achievement (i.e., one's high grades and educational experience at a prestigious university or graduate school). Some studies examine the relationship between cultural capital and competencies. For example, Fuchs and Wößmann [18] reveal that the number of books at home at the age of 16 and the his/her father's education positively influence students' literacy and numeracy. In addition, Lee and Choi [19] find that educational attainment has a positive effect on the problem-solving skills of Korean male employees. The results of previous studies suggest that adults' competencies, including problem-solving skills, can vary with their cultural capital.

### 1.2. Participation in Training

Participation in training helps employees upgrade the skills that are particularly important for their environment [20]. According to Desjardins and Rubenson [11], participation in learning and educational programs leads to higher cognitive skills, which is a key competency required to stay updated within changing environments.

Participation in training can be divided into formal and informal adult education and training (AET). Formal learning is defined as educational activities in structured situations rewarded by recognized qualifications from educational or training institutions [21]. Meanwhile, informal learning refers to institutionalized and organized education that does not bring recognized, formal

qualifications [22]. Previous studies examining participation in training report a positive relationship between an adult's competencies and AET. Specifically, active research has been conducted on the relationship between informal AET and competence from the perspective of lifelong learning. For example, Choi and Kim [23] report that participation in informal AET positively affects literacy even if all other conditions are controlled.

Participation in training can also be divided into job- and non-job-related topics. Choi [24] reports that literacy and numeracy are positively related to participation in training, and that participation in job-related training results in greater improvement than that in non-job-related training. We can infer that participating in training has a positive relationship with problem-solving skills. Furthermore, the influence of participation in training on PSTRE may vary depending on whether the training is related to the job.

### 1.3. Skill Usage

The OECD [12] highlights that the skill usage of employees is related to national economic performance and employees' success in the labor market. To emphasize that the use of skills can affect skill level, Salthouse [25] proposes the "use-it-or-lose-it" hypothesis, which suggests that staying mentally active will help maintain one's level of cognitive functioning and possibly prevent cognitive decline. Thus, those who continuously employ their skills may continue to develop their potential, while those who do not continuously employ their skills risk losing them. Some studies support this hypothesis, reporting that active skill usage prevents declines in individuals' skill level. Levels and van der Velden [26] support this assertion, especially for older people. Hultsch et al. [27] examine the hypothesis that maintaining intellectual activity mitigates the cognitive decline in later life.

Specific contexts can affect problem-solving skills. For example, regarding ICT skills, the nature of problems that people deal with differs depending on whether they are at home or work, and the place of skill usage can affect problem-solving skills. For the comprehensive development of PSTRE, it is therefore important to use skills in daily life not only at work but also at home. For example, in the context of Europe, Hämäläinen et al. [7] report that ICT skill use, both at work and at home, has a positive impact on PSTRE. In the context of the workplace culture, PSTRE can be determined by the use of influencing skills. Collaborative working is known to boost employees' skill use in the workplace [28,29]. Thus, it can be inferred that using skills in various contexts can be particularly efficient in maintaining and enhancing PSTRE.

### 1.4. Age

Seo and Kwon [30] use PIAAC data to analyze the factors affecting the skill usage of Korean employees. They find that most skills including problem-solving, ICT, and influence are negatively correlated with age, and that skill usage has been found to continuously decline after the age of 25. Moreover, as age increases, learning opportunities decline not only in relevance, but also in frequency. These results suggest that skill usage and participation in training can affect problem-solving skills with relation to age. It is therefore necessary to determine whether the interactions between age and skill usage, participation in training, and problem-solving skills are important.

### 1.5. Research Questions

Accordingly, we posed the following research questions:

RQ1: Which factors affect employees' problem-solving in TRE in Japan and Korea?

RQ2. Does the relationship between (a) participation in training and (b) skill usage and problem-solving skill in TRE vary with age? Is there any interaction between (a) and (b) and age that influences problem solving in TRE?

## 2. Data Sources and Methods

To investigate and compare the antecedents of PSTRE in Japan and Korea, we used the PIAAC 2011–2012 data. The PIAAC 2011–2012 data were collected from adults between the ages of 16 and 65 in 24 countries. It measures adults' proficiency in key information-processing skills including literacy, numeracy, and PSTRE. This survey used a direct assessment method for adult competence that facilitated objective scoring rather than using subject assessments like self-reports. In addition, we adopted an item response theory approach, in which question difficulty is adjusted according to individual competence. We also collected related information such as educational qualifications, work experience, use of skills at work and at home, work-related training, personal characteristics, background, and outcomes [12]. The current research used data from 3307 Japanese and 4540 Korean respondents, which contain their PSTRE scores. In addition, the sample was limited to employees in the AET population, which excluded PIAAC respondents aged 16 to 24. After this exclusion, the sample analyzed in this research consists of 1572 Japanese and 1637 Korean respondents.

Because of the complex sampling design of the PIAAC survey, weights were assigned to adjust for estimates of nationally representative figures [29]. The PIAAC provides a set of 10 plausible values or statistically estimated PSTRE scores. To facilitate the meaningful interpretation of results, the PIAAC suggests proficiency levels and specific score cutoffs. As shown in Table 1, PSTRE scores were classified into four levels [12]. Figure 1 shows the percentage of those aged 16–65 for each level.

**Table 1.** Description of proficiency levels in problem-solving skills in technology-rich environments (PSTRE).

| Proficiency Level | Score | Types of Tasks Completed Successfully at Each Level |
|---|---|---|
| Below level 1 | Below 241 | Tasks are based on well-defined problems involving the use of only one function within a generic interface to meet one explicit criterion without any categorical or inferential reasoning or transforming of information. Few steps are required, and no sub-goal has to be generated. |
| 1 | 241 to less than 291 | Tasks typically require the use of widely available and familiar technology applications, such as e-mail software or a web browser. There is little or no navigation required to access the information or commands required to solve the problem. The problem may be solved regardless of the respondent's awareness and use of specific tools and functions (e.g., a sort function). The tasks involve few steps and a minimal number of operators. At the cognitive level, the respondent can readily infer the goal from the task statement; problem resolution requires the respondent to apply explicit criteria; and there are few monitoring demands. |
| 2 | 291 to less than 341 | Tasks typically require the use of both generic and more specific technology applications (e.g., a novel online form). Some navigation across pages and applications is required to solve the problem. The use of tools (e.g., a sort function) can facilitate the resolution of the problem. The task may involve multiple steps and operators. The goal of the problem may have to be defined by the respondent, though the criteria to be met are explicit. There are higher monitoring demands. |
| 3 | Equal to or higher than 341 | Tasks typically require the use of both generic and more specific technology applications. Some navigation across pages and applications is required to solve the problem. The use of tools (e.g., a sort function) is required to make progress toward the solution. The task may involve multiple steps and operators. The goal of the problem may have to be defined by the respondent, and the criteria to be met may or may not be explicit. There are typically high monitoring demands. |

Source: OECD. (2013). OECD skills outlook 2013: First results from the survey of adult skills. Retrieved from http://dx.doi.org/10.1787/9789264204256-en.

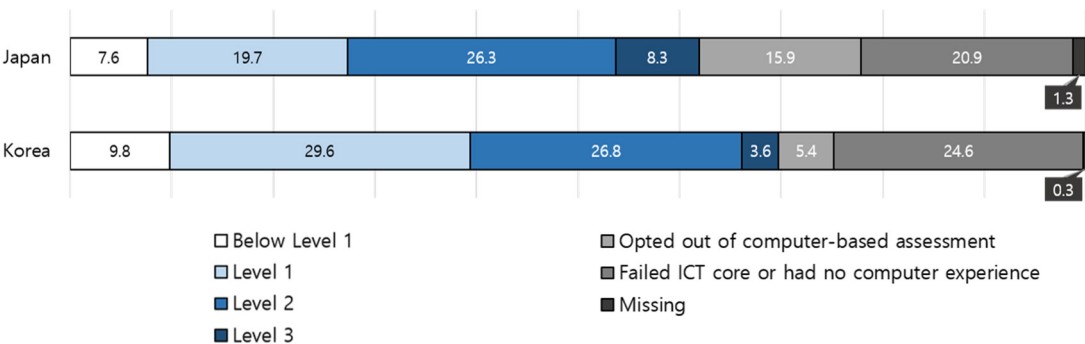

**Figure 1.** Proficiency in PSTRE among adults in Japan and Korea.

The variables used in the research are presented in Table 2. The dependent variable is PSTRE, defined as "using digital technology, communication tools, and networks to acquire and evaluate information, communicate with others, and perform practical tasks" in PIAAC. In other words, the focus is not on the use of ICT tools and applications themselves, but on the capacity to use the tools to access, process, evaluate, and analyze information effectively [12]. The independent variables include cultural capital, participation in training, and skill usage. Cultural capital was measured based on the number of books at the respondent's home at age 16, the education level of the respondent's father or male guardian, and the respondent's own education attainment. Participation in training measures whether the respondents had participated in education and training in the 12 months prior to the interview and was divided into four variables: Job-related, non-job-related, formal, and informal. Five indicators of skill usage were used: Influence, simple or complex problem-solving, and ICT skill use at work and at home. Following the PIAAC, we measured skill use based on the frequency of application. To investigate research question 2, age was used as a moderating variable. Demographics including gender, occupation, tenure, working hours, and monthly wage were controlled.

**Table 2.** Description of variables.

| Type | Variable | | Values |
|---|---|---|---|
| Dependent | PSTRE | | 0–500 |
| Independent | Cultural Capital | Number of books | 1 = 10 or less<br>2 = 11–50<br>3 = 51–259<br>4 = 251–1000<br>5 = 1000+ |
| | | Father's education | 1 = ISCED 1, 2, AND 3C short<br>2 = ISCED 3(excluding 3C short) and 4 |
| | | Education attainment | 1 = lower secondary or less<br>2 = upper secondary<br>3 = post-secondary (non-tertiary)<br>4 = professional degree<br>5 = bachelor's degree<br>6 = master's/research degree |
| | Participation in training | Formal AET for job-related reasons | 0 = did not participate<br>1 = participated |
| | | Informal AET for job-related reasons | |
| | | Formal AET for non-job-related reasons | |
| | | Informal AET for non-job-related reasons | |

**Table 2.** *Cont.*

| Type | Variable | Values |
|---|---|---|
| Skill usage | Use of influencing skills | 1 = up to 20%<br>2 = 20–40%<br>3 = 40–60%<br>4 = 60–80%<br>5 = more than 80%<br>(categorized WLE) |
| | Solving simple problems | 1 = never<br>2 = less than once a month<br>3 = less than once a week but at least once a month<br>4 = at least once a week but not every day |
| | Solving complex problems | 5 = every day |
| | Use of ICT skills at work | 1 = lowest to 20%<br>2 ≥ 20% to 40%<br>3 ≥ 40% to 60%<br>4 ≥ 60% to 80%<br>5 ≥ 80% |
| | Use of ICT skills at home | (categorized WLE) |
| Moderating | Age | Respondent's age |
| Control | Gender | 1 = male<br>2 = female |
| | Occupation | 1 = skilled<br>2 = semi-skilled white collar<br>3 = semi-skilled blue-collar<br>4 = elementary |
| | Tenure | Time spent in the current job |
| | Working hours | Weekly working hours |
| | Monthly wage | 1 = Lowest decile<br>2 = 9th decile<br>3 = 8th decile<br>4 = 7th decile<br>5 = 6th decile<br>6 = 5th decile<br>7 = 4th decile<br>8 = 3rd decile<br>9 = 2nd decile<br>10 = Highest decile<br>(Monthly earning including bonuses, in decile) |

We developed our models according to the research questions. First, to understand the influence of independent variables on the dependent variable, we input the independent variables and control variables into the model. In our sample, the age of employees ranged from 25 to 65. Age squared was included to examine whether problem-solving ability increases with age but decreases after a specific point. Second, we added the interaction terms of the variables of participation in training and skill usage and age to investigate the moderating effects. The data were analyzed using IDB Analyzer and SPSS 23.0. IDB Analyzer software, which is available for download through the PIAAC gateway website, allowed for linear regression means comparisons of plausible values for PSTRE using sample weights [31]. Table 3 reports the respondent characteristics used in this research.

**Table 3.** Characteristics of the Japanese and Korean respondents.

| | Japan n = 1572 | | Korea n = 1637 | |
|---|---|---|---|---|
| | Mean | SD | Mean | SD |
| PSTRE | 303.14 | 1.53 | 287.88 | 1.32 |
| Basic background | | | | |
| Age | 41.14 | 9.95 | 37.82 | 8.63 |
| Age squared | 179.11 | 85.73 | 150.48 | 69.92 |
| Working hour | 43.69 | 12.66 | 44.38 | 13.82 |
| Monthly wage | 6.7 | 2.58 | 6.53 | 2.53 |
| Tenure | 12.29 | 10.58 | 6.22 | 7.93 |
| Female (%) | 36 | 0.48 | 39 | 0.49 |
| ISCOSKIL4_D2(%) | 33 | 0.47 | 39 | 0.49 |
| ISCOSKIL4_D3(%) | 13 | 0.34 | 12 | 0.32 |
| ISCOSKIL4_D4(%) | 1 | 0.1 | 3 | 0.17 |
| Cultural capital | | | | |
| Books at home | 3.18 | 1.29 | 3.18 | 1.21 |
| Father's education_D2(%) | 44 | 0.5 | 36 | 0.48 |
| Father's education_D3(%) | 3 | 0.46 | 18 | 0.39 |
| EDCAT6_D2(%) | 29 | 0.46 | 3 | 0.46 |
| EDCAT6_D3(%) | 2 | 0.13 | 0 | 0 |
| EDCAT6_D4(%) | 2 | 0.4 | 25 | 0.43 |
| EDCAT6_D5(%) | 38 | 0.48 | 37 | 0.48 |
| EDCAT6_D6(%) | 7 | 0.26 | 7 | 0.26 |
| Participation in training | | | | |
| FAET12JR_D2(%) | 2 | 0.14 | 6 | 0.24 |
| FAET12NJR_D2(%) | 1 | 0.07 | 2 | 0.13 |
| NFE12JR_D2(%) | 57 | 0.49 | 65 | 0.48 |
| NFE12NJR_D2(%) | 5 | 0.21 | 11 | 0.31 |
| Skill usage | | | | |
| Use of ICT skills at home | 2.25 | 1.22 | 2.88 | 1.43 |
| Use of ICT skills at work | 2.7 | 1.46 | 3.21 | 1.56 |
| Use of influencing skills | 2.96 | 1.41 | 3.1 | 1.44 |
| Solving simple problems | 3.78 | 1.15 | 3.69 | 1.25 |
| Solving complex problems | 2.85 | 1.09 | 3 | 1.16 |

## 3. Results

Table 4 and Figure 2 summarize the means of problem-solving skill and percentage by age in Japan and Korea. For both Japan (312.66) and Korea (295.05), the 25–34 age group scores the highest. Furthermore, the Japan scores are higher than those for Korea in every age group. This is consistent with the findings of Cummins et al. [13]. It is interesting to note that the PSTRE score gap in Japan is the largest between the 35–44 and 45–54 age groups (20.23), but in Korea, there is no significant gap in the PSTRE between these two groups. In Japan, the smallest score gap is observed between the 25–34 and 35–44 age groups (7.29), while in Korea, this gap is the smallest between the 45–54 and 55 and above age groups. Figure 3 illustrates the results of PSTRE percentage by age group. In the 55 and above group, the proportions are far greater for Japan (18.05) than Korea (5.31).

**Table 4.** Summary of problem-solving skills by age group.

| | Japan | | | | | Korea | | | | |
|---|---|---|---|---|---|---|---|---|---|---|
| Age | N | % | PSTRE | Gap | Std.Dev | Age | N | % | PSTRE | Gap | Std.Dev |
| 25–34 | 568 | 28.37 | 312.66 | | 39.35 | 25–34 | 771 | 37.33 | 295.05 | | 34.67 |
| 35–44 | 707 | 34.35 | 305.37 | 7.29 | 40.36 | 35–44 | 773 | 38.43 | 280.87 | 14.18 | 34.70 |
| 45–54 | 516 | 24.45 | 285.14 | 20.23 | 42.76 | 45–54 | 403 | 18.93 | 266.61 | 14.26 | 37.62 |
| 55 and above | 290 | 12.83 | 267.09 | 18.05 | 48.27 | 55 and above | 125 | 5.31 | 259.75 | 6.86 | 34.48 |

Notes: All figures are weighted.

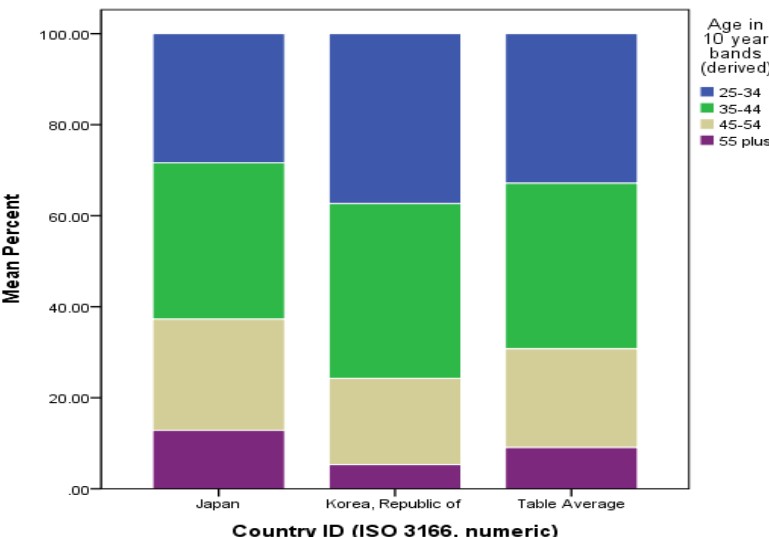

**Figure 2.** PSTRE scores by age group.

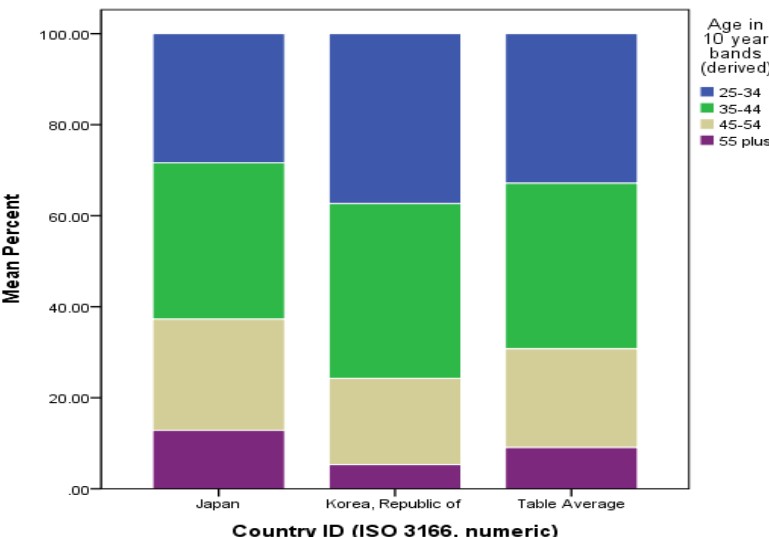

**Figure 3.** PSTRE percentage by age group.

Table 5 and Figures 4 and 5 summarize the means of problem-solving skills and percentage of ICT skill use at home. The figures for ICT skill use at home indicate similar patterns for Japan and Korea. We can see that ICT skill use at home enhances PSTRE in both countries.

**Table 5.** Summary of problem-solving skills through the use of ICT skills at home.

| Skill Use at Home | Japan | | | | Korea | | | |
| --- | --- | --- | --- | --- | --- | --- | --- | --- |
| | N | % | PSTRE | Std.Dev | N | % | PSTRE | Std.Dev |
| All zero response | 29 | 1.63 | 267.97 | 48.32 | 26 | 1.33 | 257.94 | 35.37 |
| Lowest to 20% | 627 | 33.07 | 284.37 | 44.71 | 518 | 26.19 | 265.01 | 35.20 |
| > 20% to 40% | 611 | 31.96 | 300.90 | 39.97 | 464 | 23.40 | 283.53 | 34.64 |
| > 40% to 60% | 356 | 18.11 | 312.82 | 39.40 | 344 | 17.11 | 290.91 | 35.24 |
| > 60% to 80% | 186 | 9.34 | 314.83 | 40.23 | 316 | 15.67 | 294.18 | 33.77 |
| > 80% | 112 | 5.89 | 322.86 | 43.77 | 331 | 16.3 | 295.81 | 34.77 |

Notes: All figures are weighted.

**Figure 4.** PSTRE by ICT skill use at home.

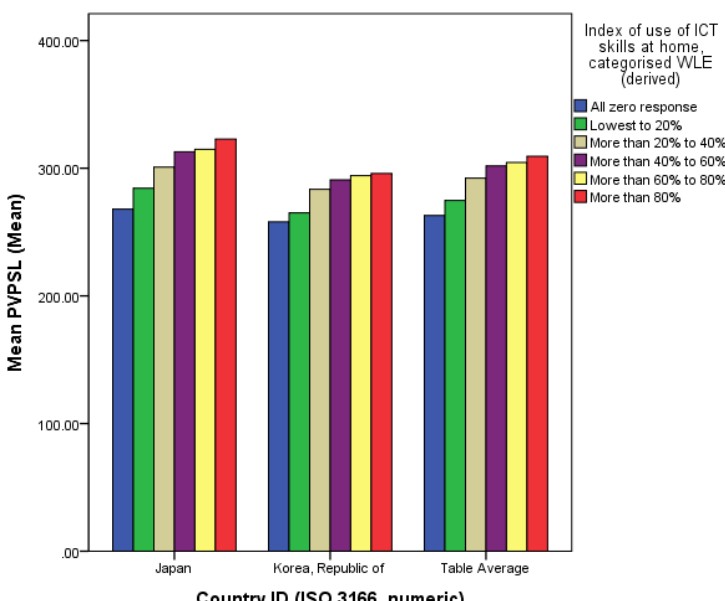

**Figure 5.** PSTRE by ICT skill use at home.

Table 6 and Figures 6 and 7 summarize the means of problem-solving skills and percentage of ICT skill use at work. There is no similarity in ICT skill use at work between the two countries. Japan (317.34) records the highest score in the 60% to 80% category and the score drops thereafter, while Korea records the highest score in the more than 80% category. Table 6 provides the results of PSTRE percentage for skill use at work. Japan records the largest percentage (22.78) in the lowest to 20% category, whereas Korea records the largest percentage (30.35) in the more than 80% category.

**Table 6.** Summary of problem-solving skills (ICT skill use at work).

| | Japan | | | | Korea | | | |
|---|---|---|---|---|---|---|---|---|
| Skill Use at Work | N | % | PSTRE | Std.Dev | N | % | PSTRE | Std.Dev |
| All zero response | 86 | 4.64 | 271.94 | 46. 08 | 54 | 3.25 | 260.98 | 46.08 |
| Lowest to 20% | 426 | 22.78 | 282.74 | 42.51 | 280 | 16.44 | 269.77 | 42.51 |
| > 20% to 40% | 380 | 20.54 | 296.31 | 39.74 | 318 | 17.81 | 279.66 | 39.74 |
| > 40% to 60% | 356 | 18.96 | 310.75 | 39.58 | 260 | 14.88 | 287.54 | 39.58 |
| > 60% to 80% | 385 | 21.32 | 317.34 | 38.03 | 302 | 17.27 | 294.72 | 39.03 |
| > 80% | 208 | 11.76 | 313.86 | 42.13 | 517 | 30.35 | 298.56 | 42.13 |

Notes: All figures are weighted.

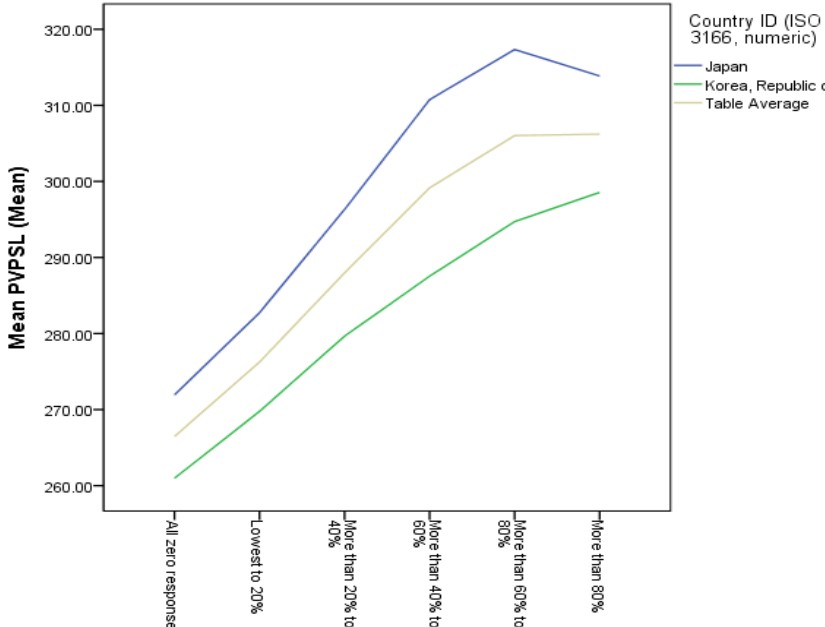

**Figure 6.** PSTRE by ICT skill use at work.

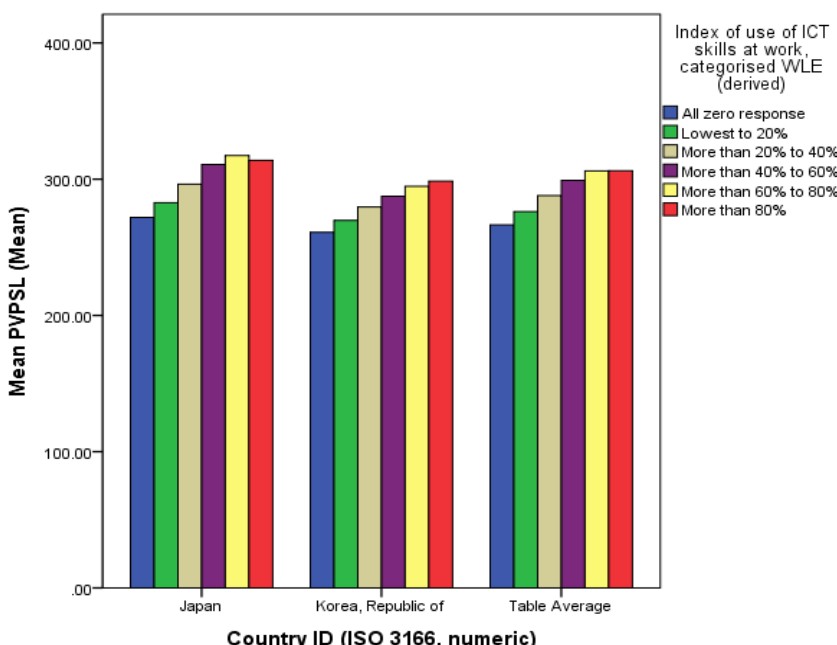

**Figure 7.** PSTRE by ICT skill use at work.

We used hierarchical ordinary least squares (OLS) regression to investigate whether aging has different impacts on the variables. We employed four clusters (basic background, cultural capital, skill use, participation in training) and focused on two clusters (use of skills and participation in training) by age. Table 7 presents the results for aging, use of skills, and participation in training for Models 1 and 3.

Model 1 in Table 7 presents the results of regressing PSTRE on the basic background cluster, the cultural capital cluster, the skill usage cluster, and the participation in training cluster in Japan. In the basic background cluster, the effect of age squared, monthly wage, and gender (female) on PSTRE were statistically significant (−0.37, 1.57, −4.91, respectively). In the cultural capital cluster, the education attainments made a significant difference to PSTRE. In the skill usage cluster, use of ICT skills at home and at work, use of influencing skills, and solving simple problems were statistically significant (6.09, 4.37, −4.6, 3.11, respectively). In the participation in training cluster, no significant difference was observed.

Model 3 in Table 7 presents the results of regressing PSTRE on the basic background cluster, the cultural capital cluster, the skill usage cluster, and the participation in training cluster in Korea. In the basic background cluster, tenure and gender (female) were statistically significant (−0.28 and −6.32, respectively). In the cultural capital cluster, the effect of books at home were statistically significant (1.79). In addition, the education attainments of respondents and their father made a significant difference to PSTRE. In the skill usage cluster, use of ICT skills at home and at work, and use of influencing skills were statistically significant (1.84, 2.31, −1.61, respectively). In the participation in training cluster, no significant difference was observed.

The results of Model 2 and Model 4 indicate the interaction effect of the use of ICT skills at home and age in Japan and Korea: The positive effect of the use of ICT skills at home on PSTRE becomes more evident as employees get older (B = 0.23, $p < 0.01$). Meanwhile, there is no significant result from Model 4.

**Table 7.** Estimation results for problem-solving skills.

| | Japan n = 1572 | | | | Korea n = 1637 | | | |
|---|---|---|---|---|---|---|---|---|
| | Model 1 | | Model 2 | | Model 3 | | Model 4 | |
| | **B** | **S.E.** | **B** | **S.E.** | **B** | **S.E.** | **B** | **S.E.** |
| Basic background | | | | | | | | |
| Age | 1.58 | 1.06 | 1.29 | 1.25 | −1.4 | 1.01 | −1.25 | 1.24 |
| Age squared | −0.37 *** | 0.12 | −0.36 *** | 0.13 | 0.04 | 0.12 | 0.04 | 0.13 |
| Working hours | −0.06 | 0.11 | −0.06 | 0.11 | 0.11 | 0.07 | 0.1 | 0.07 |
| Monthly wage | 1.57 ** | 0.69 | 1.62 ** | 0.7 | 0.63 | 0.54 | 0.62 | 0.53 |
| Tenure | 0.15 | 0.15 | 0.18 | 0.14 | −0.28 * | 0.15 | −0.32 ** | 0.15 |
| Female | −4.91 * | 2.81 | −4.77 * | 2.77 | −6.32 ** | 2.38 | −6.43 *** | 2.4 |
| ISCOSKIL4_D2 | −1.8 | 2.64 | −1.97 | 2.65 | −4.3 * | 2.21 | −4.25 * | 2.2 |
| ISCOSKIL4_D3 | −3.2 | 4.32 | −3.19 | 4.35 | −8.33 ** | 4.01 | −8.29 ** | 3.94 |
| ISCOSKIL4_D4 | −10.03 | 10.37 | −9.49 | 10.09 | −6.8 | 5.85 | −6.94 | 5.77 |
| Cultural capital | | | | | | | | |
| Books at home | 1.83 | 1.14 | 1.88 | 1.15 | 1.79 ** | 0.84 | 1.87 ** | 0.85 |
| Father's Education_D2 | −2.72 | 2.89 | −3.19 | 2.92 | 1.73 | 2.17 | 1.64 | 2.17 |
| Father's Education_D3 | 3.51 | 3.1 | 2.91 | 3.2 | 6.03 ** | 3.04 | 6.02 ** | 3.05 |
| EDCAT6_D2 | 11.61 * | 6.68 | 11.83 * | 6.7 | 9.96 | 8.73 | 9.99 | 8.82 |
| EDCAT6_D3 | 5.23 | 9.98 | 5 | 10.05 | | | | |
| EDCAT6_D4 | 15.88 *** | 6.11 | 16.15 *** | 6.1 | 20.26 ** | 8.74 | 20.57 ** | 8.84 |
| EDCAT6_D5 | 23.05 *** | 6.86 | 23.49 *** | 6.88 | 23.59 *** | 8.65 | 24.12 *** | 8.74 |
| EDCAT6_D6 | 29.45 *** | 7.51 | 30.3 *** | 7.64 | 26.88 *** | 9.73 | 27.32 *** | 9.75 |
| Skill usage | | | | | | | | |
| Use of ICT skills at home | 6.09 *** | 0.9 | −3.66 | 3.72 | 1.84 ** | 0.72 | 1.81 | 3.94 |
| Use of ICT skills at work | 4.37 *** | 0.99 | 8.12 ** | 3.31 | 2.31 *** | 0.8 | 2.59 | 3.31 |
| Use of influencing skills | −4.6 *** | 1.09 | −1.31 | 3.48 | −1.61 ** | 0.8 | 0.21 | 3.6 |
| Solving simple problems | 3.11 *** | 1.11 | 4.07 | 4.4 | 1.28 | 0.83 | 1.64 | 4.45 |
| Solving complex problems | 0.78 | 1.43 | −0.37 | 5.16 | 0.68 | 1.06 | 1.69 | 4.95 |
| Participation in training | | | | | | | | |
| FAET12JR_D2 | −4.37 | 6.86 | 13.33 | 28.23 | 4.92 | 5.1 | 2.92 | 16.99 |
| FAET12NJR_D2 | 19.06 | 14.39 | −3.42 | 32.33 | −4.66 | 7.57 | 21.8 | 29.86 |
| NFE12JR_D2 | −1.11 | 2.37 | −10.58 | 10.25 | 0.26 | 2.38 | −8.57 | 10.89 |
| NFE12NJR_D2 | 3.4 | 5.57 | −10.4 | 22.82 | 6.32 * | 3.41 | −17.37 | 16.71 |
| Interaction | | | | | | | | |
| COM * AGE | | | 0.03 | 0.12 | | | 0.03 | 0.12 |
| SIM * AGE | | | −0.02 | 0.1 | | | −0.01 | 0.11 |
| INFLU * AGE | | | −0.08 | 0.08 | | | −0.05 | 0.09 |
| ICTHOME * AGE | | | 0.23 *** | 0.09 | | | 0.00 | 0.11 |
| ICTWORK * AGE | | | −0.09 | 0.08 | | | −0.01 | 0.09 |
| FAET * AGE | | | −0.42 | 0.63 | | | 0.06 | 0.47 |
| FAETNJR * AGE | | | 0.6 | 0.65 | | | −0.69 | 0.79 |
| NFEJR * AGE | | | 0.23 | 0.24 | | | 0.23 | 0.29 |
| NFENJR * AGE | | | 0.34 | 0.5 | | | 0.62 | 0.45 |
| (CONSTANT) | 248.2 *** | 22.5 *** | 258.41 *** | 30.13 | 291.33 *** | 23.27 *** | 286.87 *** | 30.83 |
| *R square* | 0.32 | | 0.33 | | 0.28 | | 0.29 | |

Notes: All figures are weighted. Age squared is divided by 100. *, **, and *** indicate significance at the 10%, 5%, and 1% levels, respectively.

## 4. Discussion

The results reveal statistically significant differences between Japan and Korea. In the basic background cluster, the age squared has negative effects on PSTRE in Model 1, similar to previous studies [7,9,32,33], and occupation has negative effects on PSTRE in Model 3, in line with Tikkanen and Nissinen [34]. Japan's PSTRE score decreases with age, but Korea's scores differ depending on the type of job. This suggests that white collar occupations in Korea may have managed to preserve employees' PSTRE levels, but no such effect of occupation is seen in Japan. These variations in PSTRE scores by occupation imply that other variables may affect PSTRE besides occupation.

In the cultural capital cluster, the number of books at home at age 16 and education level have a positive effect on PSTRE in Models 1 and 3. Interestingly, the number of books has a long-term positive impact on PSTRE, lasting up to adulthood. Based on the analysis results, the effects of reading and schooling, as means of gaining social capital, cannot be accumulated over a short period [7,14,35], but it can be inferred that they increase problem-solving scores over an extended period [7,9,17].

In the skill usage cluster, the use of ICT skills at work and home has positive effects on PSTRE. The use of ICT skills at work and home directly enhances PSTRE through the continuous use of technology; this is in line with the "use-it-or-lose-it" hypothesis in Barrett and Readel [36–38]. However, the use of influencing skills negatively affects PSTRE. Although individuals with competent problem-solving skills are considered the best leaders to manage and supervise their subordinates [39], the analysis of the skill usage cluster reveals that leadership is not necessarily related to cognitive problem-solving ability. This provides insights into individuals' direct and indirect skill use and their relationship with PSTRE.

In the participation in training cluster, formal and informal AET for job- and non-job-related reasons do not have a significant effect on PSTRE. This result coincides with the notion that lower PSTRE is more related to socio-demographic factors (e.g., age, occupation, and gender) and work-related and everyday life skill usage factors (e.g., ICT skill use at work and home) than vocational education and training [40]. However, caution should be exercised when interpreting these results because of the relatively low number of observations for this cluster [34]. While previous studies stress employees' learning and training in the workplace [31,41–43], the ratio of Japanese respondents' participation in training to that of Korean respondents is very low. Therefore, it seems reasonable to infer that training has no effect on PSTRE due to the low participation rate. Patterson [31] demonstrates that most adults (90%) do not avail education themselves due to their expenditure on their children's education, low income, and work and family responsibilities. Situational factors such as health, disability, low social trust, and difficulty in learning new ideas, and institutional factors such as training costs and low work schedule flexibility have also been highlighted as factors that hinder learning [31].

After evaluating the impact of the basic background, cultural capital, skill usage, and participation in training clusters, we added an aging effect interaction in Models 2 and 4. We expected that the skill usage cluster (use of ICT skills at home, use of ICT skills at work, use of influencing skills, solving simple problems, solving complex problems) and the participation in training cluster (formal and informal AET for job- and non-job-related reasons) would have interactive effects on PSTRE by age. The results of Model 2 indicate the interaction effect of the use of ICT skills at home and age: The positive effect of the use of ICT skills at home on PSTRE becomes more evident as employees get older (B = 0.23, $p < 0.01$). Meanwhile, there is no significant result from Model 4. This finding shows that the use of skills at home can prevent skills loss, especially for older people in Japan. In addition, based on the "use-it-or-lose-it" perspective, the use of skills at the workplace, where employees follow established rules, as well as the use of ICT skills at home in their daily lives, are important factors in maintaining and enhancing PSTRE levels [26].

## 5. Conclusions

PSTRE is one of the important factors influencing sustainable employability in a rapidly changing workplace environment through cognitive ability and adaptability. This research compared four clusters to determine the factors affecting PSTRE in Japan and Korea. In addition, we examined whether aging moderate the relationship between skill usage and participation in training and PSTRE. Our results provide the theoretical and practical contributions for vocational psychology and work-life research in the context of employers, employees, as well as policymakers.

The implications are as follows. First, skill usage is the most effective factor in improving problem solving and coping with skill degradation. Especially, both work-related and everyday life ICT skill usage is positively related to PSTRE in Japan and Korea. This result provides supporting evidence

for the enrichment theory in terms of the work-life domain in vocational psychology [43–46] and the use-it-or-lose-it theory [36–38]

Second, we found that participation in training was not significantly related to PSTRE. These results might be attributed to the participation rate in formal and nonformal training. According to the PIAAC report [30], Japan and Korea show the lowest participation rate in formal education and training among OECD countries. The results should be considered in the context of the low participation in education and training in Japan and Korea. On the other hand, the result may be due to reverse causality. Although the vocational education and training is regarded as an important factor [11,20], some researchers [40] reported that participation in education and training was negatively related to PSTRE. They argued that employees who lacked the PSTRE were likely to have more need of education and training. Thus, caution should be exercised when interpreting these results.

Third, our research's strength lies in its large sample and the discovery of the commonalities and differences among the factors enhancing PSTRE in Japan and Korea. However, a possible limitation is that our research analyzed employees only in Japan and Korea. The analysis needs to be expanded to other OECD countries to generalize our findings. Although Japan and Korea are geographically close, our results highlight differences in the factors affecting PSTRE. Future studies should clarify whether these differences are based on national-level differences. In addition, our study also has the following limitation. In measuring PSTRE, the PIAAC excluded those who opted out of computer-based assessment, failed the ICT core test, or had no computer experience. Hence, our results cannot be generalized to all employees in Japan and Korea.

Finally, the recent rapid spread of the COVID-19 has drastically reduced personal contact. Many countries have suspended school and cancelled various meetings to secure social distancing and minimize potential damage. Furthermore, many activities that were undertaken through personal contact are being replaced by online engagement in organizations. Since it is possible that disasters such as the COVID-19 outbreak can re-occur, ICT skills for online work will become even more important, and middle-aged employees will need to raise their ICT capabilities beyond current levels; continuous use of ICT technology can prevent the PSTRE from declining with age. Otherwise, they will face difficulties not only in their current jobs, but also in re-employment.

**Author Contributions:** Conceptualization, C.-Y.J. and Y.L; methodology, S.P.; software, Y.L., S.P. and E.C.; writing—original draft preparation, S.P., E.C. and R.C.; writing—review and editing, Y.L.; supervision, C.-Y.J.; project administration, C.-Y.J. All authors have read and agreed to the published version of the manuscript.

**Funding:** This research received no external funding.

**Conflicts of Interest:** The authors declare no conflict of interest.

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
