# Peer review of "Factors Affecting Employees’ Problem-Solving Skills in Technology-Rich Environments in Japan and Korea"

_sustainability, doi:10.3390/su12177079_

Round 1

Reviewer 1 Report

Dear authors, your article is interesting but I need you to answer some questions:

INTRODUCTION and LITERATURE REVIEW

  • “Introduction” and “literature review” should form a single section.
  • “Literature review” should not be to have subsections.

DATA SOURCES AND METHODS

Design, sample:

Authors must specify the research design.

What was the target population? How was the sample chosen? Authors must specify it.

Study, site, population

The authors must include the response rate of the participants in the study.

Ethical considerations:

Have you consulted the ethics committee? Authors must mention and say the reference.

DATA ANALYSIS and ESTIMATION RESULTS

“Data analysis” and “estimation results” should go together in a single section called "Results".

DISCUSSION and CONCLUSIONS

The discussion is very short. Authors should better analyze their results (they are very extensive).

REFERENCES

Many bibliographies are obsolete and some citations are incomplete. The bibliographic citations used are more than 5 years old (37%). The authors must update and arrange the bibliography.

Author Response

  1. We combined "Introducion" and "literature review" in a single section, "Introduction" as below(pp. 2-4).
    1. Introduction

    The latest industrial revolution, Industry 4.0, has led to significant reforms in the economic and social environment with regard to information and communication technology (ICT), thereby transforming individuals’ lives. Due to technological developments, computers and other ICT tools have been integrated into daily lives [1], and it has become necessary for individuals to be able to use these tools effectively. In the workplace, automation has substituted routine tasks through the use of digital technologies, and employees are increasingly required to perform non-routine tasks that computers cannot easily perform [2]. Jobs requiring problem-solving skills for abstract tasks are expected to increase [3,4]. Hence, these changes create the need for problem-solving skills in technology-rich environments (PSTRE), entailing “abilities to solve problems for personal, work, and civic purposes by setting up appropriate goals and plans, and accessing and making use of information through computers and computer networks” [5]. Consequently, in today’s rapidly changing environment, PSTRE are among the most essential employee competencies [6].

    With the growing importance of PSTRE, it has become necessary to shed light on the factors influencing such skills. Previous studies have identified socio-demographic (e.g., gender, age, immigrant status, education attainment), family-related (e.g., parents’ education attainment, the number of books at home, skill usage at home), and work-related factors (e.g., PT at work, skill usage at work, job characteristics) influencing PSTRE [1,7-9]. However, research on the linkage between learning and skill development and PSTRE is scare. Skills can be converted from accumulated cultural capital within individuals’ sociocultural contexts through pedagogic action [10] or acquired directly through participation in training [11]. Learning by doing is  known to play an important role in skill development. Against this backdrop, the current research investigates the relative impact of the following factors on PSTRE: cultural capital, participation in training, and skill usage. Additionally, since participation in training and skill usage are lifelong processes, the interaction effect between these two factors and age is analyzed.

    This research explored PSTRE and its factors with regard to employees in Japan and Korea. Although Japan and Korea have geographic, societal, and cultural similarities, the level of skills differs between these two countries [12]. A cross-country comparative view may help identify the commonalities and differences among the factors enhancing problem solving in TRE in Japan and Korea. This research used data from the Programme for the International Assessment of Adult Competencies (PIAAC). The PIAAC data provide information on adults’ proficiency in literacy, numeracy, and problem solving in TRE considering various background characteristics for over 40 participating countries [12]. Most related research has focused on the United States and European countries [13,1,7,14]; few studies have focused on Asian countries. This research takes a significant first step in understanding PSTRE in the context of Asia, particularly Japan and Korea, and offers critical practical implications that will be useful in improving employees’ problem-solving proficiency.

    1.1. Cultural capital

    Bourdieu [10] defines cultural capital as a mechanism of class reproduction with economic and social capital through “the form of capital.” He introduced this concept to explain the elite’s advantageous access to better schools and jobs, which is not accounted for by human capital (talent and skills) alone. According to Bourdieu [10], the attitudes, features, and behaviors of working-class students are incongruent with those of middle-class students. Students from privileged backgrounds encounter similar cultural experiences, which helps them advance in terms of the general culture, linguistic skills, knowledge, and other components of the educational system [15, 16]. On the other hand, students from disadvantageous socioeconomic backgrounds have difficulty adjusting to school and have a higher chance of failing a class. Cultural capital can have lasting effects on the life of an individual. Gustafsson [17] reports that performance differences in PIAAC are strongly related to Programme for International Student Assessment(PISA) achievement trends in literacy and numeracy, which suggests that the effect of success in school on competence can persist in the long term.

    Cultural capital exists in three forms: embodied, objectified, and institutionalized. Embodied cultural capital is the natural way of thinking, judgment, and preferences that an individual develops at home. Objectified cultural capital refers to cultural goods such as pictures, books, dictionaries, and instruments. Institutionalized cultural capital implies high educational achievement (i.e., one’s high grades and educational experience at a prestigious university or graduate school). Some studies examine the relationship between cultural capital and competencies. For example, Fuchs and Wößmann [18] reveal that the number of books at home at the age of 16 and the his/her father’s education positively influence students’ literacy and numeracy. In addition, Lee and Choi [19] find that educational attainment has a positive effect on the problem-solving skills of Korean male employees. The results of previous studies suggest that adults’ competencies, including problem-solving skills, can vary with their cultural capital.

    1.2. Participation in training

     Participation in training helps employees upgrade the skills that are particularly important for their environment [20]. According to Desjardins and Rubenson [11], participation in learning and educational programs leads to higher cognitive skills, which is a key competency required to stay updated within changing environments.

    Participation in training can be divided into formal and informal adult education and training (AET). Formal learning is defined as educational activities in structured situations rewarded by recognized qualifications from educational or training institutions [21]. Meanwhile, informal learning refers to institutionalized and organized education that does not bring recognized, formal qualifications [22]. Previous studies examining participation in training report a positive relationship between an adult’s competencies and AET. Specifically, active research has been conducted on the relationship between informal AET and competence from the perspective of lifelong learning. For example, Choi and Kim [23] report that participation in informal AET positively affects literacy even if all other conditions are controlled.

    Participation in training can also be divided into job- and non-job-related topics. Choi [24] reports that literacy and numeracy are positively related to participation in training, and that participation in job-related training results in greater improvement than that in non-job-related training. We can infer that participating in training has a positive relationship with problem-solving skills. Furthermore, the influence of participation in training on PSTRE may vary depending on whether the training is related to the job.

    1.3. Skill usage

    The OECD [12] highlights that the skill usage of employees is related to national economic performance and employees’ success in the labor market. To emphasize that the use of skills can affect skill level, Salthouse [25] proposes the “use-it-or-lose-it” hypothesis, which suggests that staying mentally active will help maintain one’s level of cognitive functioning and possibly prevent cognitive decline. Thus, those who continuously employ their skills may continue to develop their potential, while those who do not continuously employ their skills risk losing them. Some studies support this hypothesis, reporting that active skill usage prevents declines in individuals’ skill level. Levels and van der Velden [26] support this assertion, especially for older people. Hultsch et al. [27] examine the hypothesis that maintaining intellectual activity mitigates the cognitive decline in later life.

            Specific contexts can affect problem-solving skills. For example, regarding ICT skills, the nature of problems that people deal with differs depending on whether they are at home or work, and the place of skill usage can affect problem-solving skills. For the comprehensive development of PSTRE, it is therefore important to use skills in daily life not only at work but also at home. For example, in the context of Europe, Hämäläinen et al. [7] report that ICT skill use, both at work and at home, has a positive impact on PSTRE. In the context of the workplace culture, PSTRE can be determined by the use of influencing skills. Collaborative working is known to boost employees’ skill use in the workplace [28,29]. Thus, it can be inferred that using skills in various contexts can be particularly efficient in maintaining and enhancing PSTRE.

    1.4. Age

    Seo and Kwon [30] use PIAAC data to analyze the factors affecting the skill usage of Korean employees. They find that most skills including problem-solving, ICT, and influence are negatively correlated with age, and that skill usage has been found to continuously decline after the age of 25. Moreover, as age increases, learning opportunities decline not only in relevance, but also in frequency. These results suggest that skill usage and participation in training can affect problem-solving skills with relation to age. It is therefore necessary to determine whether the interactions between age and skill usage, participation in training, and problem-solving skills are important.

    1.5. Research questions

    Accordingly, we posed the following research questions:

    RQ1: Which factors affect employees’ problem-solving in TRE in Japan and Korea?

    RQ2. Does the relationship between (a) participation in training and (b) skill usage and problem-solving skill in TRE vary with age? Is there any interaction between (a) and (b) and age that influences problem solving in TRE?

  2. We specified the target population and the sample chosen as below(p. 4).

    To investigate and compare the antecedents of PSTRE in Japan and Korea, we used the PIAAC 2011–2012 data. The PIAAC 2011-2012 data were collected from adults between the ages of 16 and 65 in 24 countries. It measures adults’ proficiency in key information-processing skills including literacy, numeracy, and PSTRE. This survey used a direct assessment method for adult competence that facilitated objective scoring rather than using subject assessments like self-reports. In addition, we adopted an item response theory approach, in which question difficulty is adjusted according to individual competence. We also collected related information such as educational qualifications, work experience, use of skills at work and at home, work-related training, personal characteristics, background, and outcomes [12]. The current research used data from 3,307 Japanese and 4,540 Korean respondents which contains their PSTRE scores. In addition, the sample was limited to employees in the AET population, which excluded PIAAC respondents aged 16 to 24. After this exclusion, the sample analyzed in this research consists of 1,572 Japanese and 1,637 Korean respondents.

  3. We combined "Data analysis" and "estimation results" in a single section, "Results" as below(pp. 2-4)
    1. Results

    Table 4 and Figure 2 summarize the means of problem-solving skill and percentage by age in Japan and Korea. For both Japan (312.66) and Korea (295.05), the 25–34 age group scores the highest. Furthermore, the Japan scores are higher than those for Korea in every age group. This is consistent with the findings of Cummins et al [13]. It is interesting to note that the PSTRE score gap in Japan is the largest between the 35–44 and 45–54 age groups (20.23), but in Korea, there is no significant gap in the PSTRE between these two groups. In Japan, the smallest score gap is observed between the 25–34 and 35–44 age groups (7.29), while in Korea, this gap is the smallest between the 45–54 and 55 and above age groups. Figure 3 illustrates the results of PSTRE percentage by age group. In the 55 and above group, the proportions are far greater for Japan (18.05) than Korea (5.31). (...)

  4. We reorganized our text in order to divide "Discussion and Conclusion" to "Discussion" and "Conclusion" and made clear the discussion of this research as below(pp. 12-14).
    1. Discussion

    The results reveal statistically significant differences between Japan and Korea. In the basic background cluster, the age squared has negative effects on PSTRE in Model 1, similar to previous studies [32,33,7,9], and occupation has negative effects on PSTRE in Model 3, in line with Tikkanen and Nissinen [34]. Japan’s PSTRE score decreases with age, but Korea’s scores differ depending on the type of job. This suggests that white collar occupations in Korea may have managed to preserve employees’ PSTRE levels, but no such effect of occupation is seen in Japan. These variations in PSTRE scores by occupation imply that other variables may affect PSTRE besides occupation.

    In the cultural capital cluster, the number of books at home at age 16 and education level have a positive effect on PSTRE in Models 1 and 3. Interestingly, the number of books has a long-term positive impact on PSTRE, lasting up to adulthood. Based on the analysis results, the effects of reading and schooling, as means of gaining social capital, cannot be accumulated over a short period [7,14,35], but it can be inferred that they increase problem-solving scores over an extended period [17,7,9].

    In the skill usage cluster, the use of ICT skills at work and home has positive effects on PSTRE. The use of ICT skills at work and home directly enhances PSTRE through the continuous use of technology; this is in line with the “use-it-or-lose-it” hypothesis in Barrett and Readel [36-38]. However, the use of influencing skills negatively affects PSTRE. Although individuals with competent problem-solving skills are considered the best leaders to manage and supervise their subordinates [39], the analysis of the skill usage cluster reveals that leadership is not necessarily related to cognitive problem-solving ability. This provides insights into individuals’ direct and indirect skill use and their relationship with PSTRE.

    In the participation in training cluster, formal and informal AET for job- and non-job-related reasons do not have a significant effect on PSTRE. This result coincides with the notion that lower PSTRE is more related to socio-demographic factors(e.g., age, occupation, and gender) and work-related and everyday life skill usage factors(e.g., ICT skill use at work and home) than vocational education and training [40]. However, caution should be exercised when interpreting these results because of the relatively low number of observations for this cluster [34]. While previous studies stress employees’ learning and training in the workplace [41,31,42], the ratio of Japanese respondents’ participation in training to that of Korean respondents is very low. Therefore, it seems reasonable to infer that training has no effect on PSTRE due to the low participation rate. Patterson [31] demonstrates that most adults (90%) do not avail education themselves due to their expenditure on their children’s education, low income, and work and family responsibilities. Situational factors such as health, disability, low social trust, and difficulty in learning new ideas, and institutional factors such as training costs and low work schedule flexibility have also been highlighted as factors that hinder learning [31].

    After evaluating the impact of the basic background, cultural capital, skill usage, and participation in training clusters, we added an aging effect interaction in Models 2 and 4. We expected that the skill usage cluster (use of ICT skills at home, use of ICT skills at work, use of influencing skills, solving simple problems, solving complex problems) and the participation in training cluster (formal and informal AET for job- and non-job-related reasons) would have interactive effects on PSTRE by age. The results of Model 2 indicate the interaction effect of the use of ICT skills at home and age: the positive effect of the use of ICT skills at home on PSTRE becomes more evident as employees get older (B = 0.23, p <.01). Meanwhile, there is no significant result from Model 4. This finding shows that the use of skills at home can prevent skills loss, especially for older people in Japan. In addition, based on the “use-it-or-lose-it” perspective, the use of skills at the workplace, where employees follow established rules, as well as the use of ICT skills at home in their daily lives, are important factors in maintaining and enhancing PSTRE levels [26].

    1. Conclusions

    PSTRE is one of the important factors influencing sustainable employability in a rapidly changing workplace environment through cognitive ability and adaptability. This research compared four clusters to determine the factors affecting PSTRE in Japan and Korea. In addition, we examined whether aging moderate the relationship between skill usage and participation in training and PSTRE. Our results provide the theoretical and practical contributions for vocational psychology and work-life research in the context of employers, employees, as well as policymakers.

    The implications are as follows. First, skill usage is the most effective factor in improving problem solving and coping with skill degradation. Especially, both work-related and everyday life ICT skill usage is positively related to PSTRE in Japan and Korea. This result provides supporting evidence for the enrichment theory in terms of the work-life domain in vocational psychology [44-46] and the use-it-or-lose-it theory[36-38]

     Second, we found that participation in training was not significantly related to PSTRE. These results might be attributed to the participation rate in formal and nonformal training. According to the PIAAC report [30], Japan and Korea shows the lowest participation rate in formal education and training among OECD countries. The results should be considered in the context of the low participation in education and training in Japan and Korea. On the other hand, the result may be due to reverse causality. Although the vocational education and training is regarded as an important factor [11, 20], some researches [40] reported that participation in education and training was negatively related to PSTRE. They argued that employees who lacked the PSTRE were likely to have more need of education and training. Thus, caution should be exercised when interpreting these results.

    Third, our research’s strength lies in its large sample and the discovery of the commonalities and differences among the factors enhancing PSTRE in Japan and Korea. However, a possible limitation is that our research analyzed employees only in Japan and Korea. The analysis needs to be expanded to other OECD countries to generalize our findings. Although Japan and Korea are geographically close, our results highlight differences in the factors affecting PSTRE. Future studies should clarify whether these differences are based on national-level differences. In addition, our study also has the following limitation. In measuring PSTRE, the PIAAC excluded those who opted out of computer-based assessment, failed the ICT core test, or had no computer experience. Hence, our results cannot be generalized to all employees in Japan and Korea.

    Finally, the recent rapid spread of the COVID-19 has drastically reduced personal contact. Many countries have suspended school and cancelled various meetings to secure social distancing and minimize potential damage. Furthermore, many activities that were undertaken through personal contact are being replaced by online engagement in organizations . Since it is possible that disasters such as the COVID-19 outbreak can re-occur, ICT skills for online work will become even more important, and middle-aged employees will need to raise their ICT capabilities beyond current levels; Continuous use of ICT technology can prevent the PSTRE from declining with age. Otherwise, they will face difficulties not only in their current jobs but also in re-employment.

  5. We updated the bibilographic citations published within 5 years.
  1. Desjardins, R., & Ederer, P. (2015). Socio-demographic and practice-oriented factors related to proficiency in problem solving: a lifelong learning perspective. International Journal of Lifelong Education, 34(4), 468-486. http://dx.doi.org/10.1080/02601370.2015.1060027
  2. Bonekamp, L., & Sure, M. (2015). Consequences of Industry 4.0 on human labour and work organisation. Journal of Business and Media Psychology, 6(1), 33-40.
  3. Autor, D. H., & Price, Brendan. (2013). The changing task composition of the US labor market: An update of Autor, Levy, and Murnane (2003). MIT Working Paper. Retrieved from https://economics.mit.edu/files/9758
  4. (2016). Forecasting Skill Demand and Supply: Employment trend. Retrieved from http://www.cedefop.europa.eu/de/events-and-projects/projects/forecasting
  5. Organisation for Economic Co-operation and Development. (2012). Literacy, numeracy and problem solving in technology-rich environments: Framework for the OECD Survey of Adult Skills. doi: 10.1787/9789264128859-en
  6. World Economic Forum. (2016). The future of jobs: Employment, skills and workforce strategy for the fourth industrial revolution. Geneva, Switzerland: World Economic Forum.
  7. Hamalainen, R., De Wever, B., Nissinen, K., & Cincinnato, S. (2019). What makes the difference - PIAAC as a resource for understanding the problem-solving skills of Europe's higher-education adults. Computers & Education, 129, 27-36. https://doi.org/10.1016/j.compedu.2018.10.013 
  8. Strakova, J., & Vesely, A. (2019). Other Things Being Equal: Comparing Literacies in the Czech and Slovak Republics. Comparative Education Review, 63(3), 418-438. http://dx.doi.org/10.1086/703869
  9. Zahavi, M., BenDavid-Hadar, I., & Klein, J. (2019). Education and competencies within Belt and Road countries. International Journal of Educational Management, 33(6), 1411-1430. http://dx.doi.org/10.1108/IJEM-03-2019-0100
  10. Guillory, J. (2013). Cultural capital: The problem of literary canon formation. University of Chicago Press.
  11. Desjardins, R., & Rubenson, K. (2013). Participation Patterns in Adult E ducation: the role of institutions and public policy frameworks in resolving coordination problems. European Journal of Education, 48(2), 262-280. https://doi.org/10.1111/ejed.12029 
  12. Organisation for Economic Co-operation and Development. (2013). OECD skills outlook 2013: First results from the survey of adult skills. Retrieved from http://dx.doi.org/10.1787/978926420 4256-en
  13. Cummins, P. A., Yamashita, T., Millar, R. J., & Sahoo, S. (2019). Problem-Solving Skills of the US Workforce and Preparedness for Job Automation. Adult Learning, 30(3), 111-120. https://doi.org/10.1177/1045159518818407 
  14. Storen, L. A., Lundetrae, K., & Boring, P. (2018). Country differences in numeracy skills: how do they vary by job characteristics and education levels? International Journal of Lifelong Education, 37(5), 578-597. https://doi.org/10.1080/02601370.2018.1554718 
  15. Boussanlègue, T., Biriziwè, H., Esso-Mondjonna, M., & Essè, A. (2020). Performances of Elementary Pupils in French and Mathematics and Socio-Professional Category and the Formal Education Level of Parents in Togo. International journal of educational review, 3(1), 53-62. https://doi.org/10.33369/ijer.v3i1.11843
  16. Hultqvist, E., & Lidegran, I. (2020). The use of ‘cultural capital’in sociology of education in sweden. International Studies in Sociology of Education, 1-8. https://doi.org/10.1080/09620214.2020.1785322
  17. Gustafsson, J. E. (2016). Lasting effects of quality of schooling: Evidence from PISA and PIAAC. Intelligence, 57, 66-72. http://dx.doi.org/10.1016/j.intell.2016.05.004
  18. Fuchs, T., & Wößmann, L. (2007). What accounts for international differences in student performance? A re-examination using PISA data. Empirical Economics, 32(2), 433-464. https://doi.org/10.1007/s00181-006-0087-0
  19. Lee, J. H., & Choi, S. (2015). Skills of Koreans: Empirical Analysis and Future Strategies. KDI Research Monograph, 8, 1-382.
  20. (2019). On the way to 2020: data for vocational education and training policies indicator overviews: 2019 update. Cedefop research paper, 76. http://data.europa.eu/doi/10.2801/62708
  21. Bozkurt, A., & Ucar, H. (2020). Blockchain Technology as a Bridging Infrastructure Among Formal, Non-Formal, and Informal Learning Processes. In Blockchain Technology Applications in Education(pp. 1-15). IGI Global.
  22. UNESCO Institute for Statistics. (2013). UIS glossary. Quebec, Canada: UNESCO Institute for Statistics. Retrieved from https://unevoc.unesco.org/go.php?q=TVETipedia+Glossary+A-Z&id=185
  23. Choi, S. J., Kim, S.N. (2015). A Comparative Study on the Influence of the Non-formal Lifelong Learning Participation and Usage of Skills to the Adult`s Key Competencies: Using Propensity Score Matching Method. Journal of vocational education, 34(4), 93-120.
  24. Choi, Y. J. (2015) An Analysis of the Effects of Adult Learning Participation on Competencies in OECD PIAAC: Using Propensity Score Matching. Journal of Lifelong Education, 21(4), 139-167.
  25. Hamm, J. M., Heckhausen, J., Shane, J., & Lachman, M. E. (2020). Risk of cognitive declines with retirement: Who declines and why?. Psychology and Aging. 35(3), 449–457. https://doi.org/10.1037/pag0000453
  26. Levels, M., & van der Velden, R. (2013). Use-it-or-lose-it? Explaining age-related differences in key information processing skills. Paper presented at the PIAAC invitational research conference: The importance of skills and how to assess them. Washington, DC. Retrieved November 13–15, 2017, from http://www.nvdemografie.nl/sites/ default/files/rolf_van_der_velden_-_keynote_use_it_or_lose_it.pdf.
  27. Chang, Y. H., Wu, I. C., & Hsiung, C. A. (2020). Reading activity prevents long-term decline in cognitive function in older people: evidence from a 14-year longitudinal study. International Psychogeriatrics, 1-12. https://doi.org/10.1017/S1041610220000812
  28. Van Laar, E., Van Deursen, A. J., Van Dijk, J. A., & De Haan, J. (2019). The sequential and conditional nature of 21st-century digital skills. International journal of communication, 40(3), 730–754. https://doi.org/10.1177/0143831X16656412
  29. Organisation for Economic Co-operation and Development. (2016). The survey of adult skills: Reader’s companion(2nd ed.). Paris, FR: Author. https://doi.org/10.1787/9789264258075-en 
  30. Seo, Y. J. & Kwon, Y. K. (2013). Korean Competence, Learning and Work. Sejong, KR: Korea Development Institute.
  31. Patterson, M. B. (2018). The forgotten 90% Adult nonparticipation in education. Adult Education Quarterly, 68(1), 41-62. http://dx.doi.org/10.1177/0741713617731810
  32. Ganzach, Y., & Patel, P. C. (2018). Wages, mental abilities and assessments in large scale international surveys: Still not much more than g. Intelligence, 69, 1-7. https://doi.org/10.1016/j.intell.2018.03.014 
  33. Lindberg, M., & Silvennoinen, H. (2018). Assessing the basic skills of the highly educated in 21 OECD countries: an international benchmark study of graduates' proficiency in literacy and numeracy using the PIAAC 2012 data. Comparative Education, 54(3), 325-351. http://dx.doi.org/10.1080/03050068.2017.1403676
  34. Tikkanen, T., & Nissinen, K. (2018). Drivers of job-related learning among low-educated employees in the Nordic countries. International Journal of Lifelong Education, 37(5), 615-632. http://dx.doi.org/10.1080/02601370.2018.1554720
  35. Yalcin, S. (2019). Competence Differences in Literacy, Numeracy, and Problem Solving According to Sex. Adult Education Quarterly, 69(2), 101-119. http://dx.doi.org/10.1177/0741713619827386
  36. Barrett, G. F., & Riddell, W. C. (2016). Ageing and Literacy Skills: Evidence from IALS, ALL and PIAAC(IZA Discussion Papers No. 10017). Bonn, GM: Institute for the Study of Labor (IZA). https://doi.org/10.1787/5jlphd2twps1-en 
  37. Allen, J., & van der Velden, R. (2013). Skills for the 21st century: Implications for dutch education. Higher education: Recent trends, emerging issues and future outlook, 1-40.
  38. Fernandez-de-Alava, M., Quesada-Pallares, C., & Garcia-Carmona, M. (2017). Use of ICTs at work: an intergenerational analysis in Spain. Cultura Y Educacion, 29(1), 120-150. https://doi.org/10.1080/11356405.2016.1274144 
  39. Tong, T. T., Li, H. Z., & Greiff, S. (2019). Human capital and leadership: the impact of cognitive and noncognitive abilities. Applied Economics, 51(53), 5741-5752. https://doi.org/10.1080/00036846.2019.1619022
  40. Hamalainen, R., De Wever, B., Malin, A., & Cincinnato, S. (2015). Education and working life: VET adults' problem-solving skills in technology-rich environments. Computers & Education, 88, 38-47. https://doi.org/10.1016/j.compedu.2015.04.013 
  41. Hidalgo-Cabrillana, A., Kuehn, Z., & Lopez-Mayan, C. (2017). Development accounting using PIAAC data. Series-Journal of the Spanish Economic Association, 8(4), 373-399. http://dx.doi.org/10.1007/s13209-017-0162-0
  42. Tikkanen, T. (2017). Problem-solving skills, skills needs and participation in lifelong learning in technology-intensive work in the Nordic countries. Sodobna Pedagogika-Journal of Contemporary Educational Studies, 68(4), 110-128.
  43. Dieckmann, U., Hager, G., Magnuszewski, P., & Lees, M. (2020). Human Capabilities for Systems Leadership: Disseminating Systems Thinking through Education and Training. https://doi.org/10.1787/879c4f7a-en
  44. Baral, R., & Bhargava, S. (2010). Work‐family enrichment as a mediator between organizational interventions for work‐life balance and job outcomes. Journal of managerial psychology, 25(3), 274-300. http://dx.doi.org/10.1108/02683941011023749
  45. Daniel, S., & Sonnentag, S. (2014). Work to non-work enrichment: The mediating roles of positive affect and positive work reflection. Work & Stress, 28(1), 49-66. http://dx.doi.org/10.1080/02678373.2013.872706
  46. Lee, E. S., Chang, J. Y., & Kim, H. (2011). The work–family interface in Korea: Can family life enrich work life?. The International Journal of Human Resource Management, 22(9), 2032-2053. https://doi.org/10.1080/09585192.2011.573976 

Reviewer 2 Report

Dear Authors,

Although the interest of the manuscript, some improvements are needed, namely:

  • On Section 3, Please explain clearly the type of sampling used.
  • Section 6 should be divided in two different sections:
    - Discussion
    - Conclusion
  • We don't have here a real discussion section. Authors only highlighted the research contributions.
  • Authors should reorganize their text in order to insert this results discussion section.
  • Also a Conclusion section is missing. Authors should insert it.
  • After reorganize the text, please make clear the theoretical and practical contributions of this study to the research field.

Author Response

We reorganized our text in order to divide "Discussion and Conclusion" to "Discussion" and "Conclusion" and made clear the discussion of this research as below(pp. 12-14).

  1. Discussion

The results reveal statistically significant differences between Japan and Korea. In the basic background cluster, the age squared has negative effects on PSTRE in Model 1, similar to previous studies [32,33,7,9], and occupation has negative effects on PSTRE in Model 3, in line with Tikkanen and Nissinen [34]. Japan’s PSTRE score decreases with age, but Korea’s scores differ depending on the type of job. This suggests that white collar occupations in Korea may have managed to preserve employees’ PSTRE levels, but no such effect of occupation is seen in Japan. These variations in PSTRE scores by occupation imply that other variables may affect PSTRE besides occupation.

In the cultural capital cluster, the number of books at home at age 16 and education level have a positive effect on PSTRE in Models 1 and 3. Interestingly, the number of books has a long-term positive impact on PSTRE, lasting up to adulthood. Based on the analysis results, the effects of reading and schooling, as means of gaining social capital, cannot be accumulated over a short period [7,14,35], but it can be inferred that they increase problem-solving scores over an extended period [17,7,9].

In the skill usage cluster, the use of ICT skills at work and home has positive effects on PSTRE. The use of ICT skills at work and home directly enhances PSTRE through the continuous use of technology; this is in line with the “use-it-or-lose-it” hypothesis in Barrett and Readel [36-38]. However, the use of influencing skills negatively affects PSTRE. Although individuals with competent problem-solving skills are considered the best leaders to manage and supervise their subordinates [39], the analysis of the skill usage cluster reveals that leadership is not necessarily related to cognitive problem-solving ability. This provides insights into individuals’ direct and indirect skill use and their relationship with PSTRE.

In the participation in training cluster, formal and informal AET for job- and non-job-related reasons do not have a significant effect on PSTRE. This result coincides with the notion that lower PSTRE is more related to socio-demographic factors(e.g., age, occupation, and gender) and work-related and everyday life skill usage factors(e.g., ICT skill use at work and home) than vocational education and training [40]. However, caution should be exercised when interpreting these results because of the relatively low number of observations for this cluster [34]. While previous studies stress employees’ learning and training in the workplace [41,31,42], the ratio of Japanese respondents’ participation in training to that of Korean respondents is very low. Therefore, it seems reasonable to infer that training has no effect on PSTRE due to the low participation rate. Patterson [31] demonstrates that most adults (90%) do not avail education themselves due to their expenditure on their children’s education, low income, and work and family responsibilities. Situational factors such as health, disability, low social trust, and difficulty in learning new ideas, and institutional factors such as training costs and low work schedule flexibility have also been highlighted as factors that hinder learning [31].

After evaluating the impact of the basic background, cultural capital, skill usage, and participation in training clusters, we added an aging effect interaction in Models 2 and 4. We expected that the skill usage cluster (use of ICT skills at home, use of ICT skills at work, use of influencing skills, solving simple problems, solving complex problems) and the participation in training cluster (formal and informal AET for job- and non-job-related reasons) would have interactive effects on PSTRE by age. The results of Model 2 indicate the interaction effect of the use of ICT skills at home and age: the positive effect of the use of ICT skills at home on PSTRE becomes more evident as employees get older (B = 0.23, p <.01). Meanwhile, there is no significant result from Model 4. This finding shows that the use of skills at home can prevent skills loss, especially for older people in Japan. In addition, based on the “use-it-or-lose-it” perspective, the use of skills at the workplace, where employees follow established rules, as well as the use of ICT skills at home in their daily lives, are important factors in maintaining and enhancing PSTRE levels [26].

  1. Conclusions

PSTRE is one of the important factors influencing sustainable employability in a rapidly changing workplace environment through cognitive ability and adaptability. This research compared four clusters to determine the factors affecting PSTRE in Japan and Korea. In addition, we examined whether aging moderate the relationship between skill usage and participation in training and PSTRE. Our results provide the theoretical and practical contributions for vocational psychology and work-life research in the context of employers, employees, as well as policymakers.

The implications are as follows. First, skill usage is the most effective factor in improving problem solving and coping with skill degradation. Especially, both work-related and everyday life ICT skill usage is positively related to PSTRE in Japan and Korea. This result provides supporting evidence for the enrichment theory in terms of the work-life domain in vocational psychology [44-46] and the use-it-or-lose-it theory[36-38]

 Second, we found that participation in training was not significantly related to PSTRE. These results might be attributed to the participation rate in formal and nonformal training. According to the PIAAC report [30], Japan and Korea shows the lowest participation rate in formal education and training among OECD countries. The results should be considered in the context of the low participation in education and training in Japan and Korea. On the other hand, the result may be due to reverse causality. Although the vocational education and training is regarded as an important factor [11, 20], some researches [40] reported that participation in education and training was negatively related to PSTRE. They argued that employees who lacked the PSTRE were likely to have more need of education and training. Thus, caution should be exercised when interpreting these results.

Third, our research’s strength lies in its large sample and the discovery of the commonalities and differences among the factors enhancing PSTRE in Japan and Korea. However, a possible limitation is that our research analyzed employees only in Japan and Korea. The analysis needs to be expanded to other OECD countries to generalize our findings. Although Japan and Korea are geographically close, our results highlight differences in the factors affecting PSTRE. Future studies should clarify whether these differences are based on national-level differences. In addition, our study also has the following limitation. In measuring PSTRE, the PIAAC excluded those who opted out of computer-based assessment, failed the ICT core test, or had no computer experience. Hence, our results cannot be generalized to all employees in Japan and Korea.

Finally, the recent rapid spread of the COVID-19 has drastically reduced personal contact. Many countries have suspended school and cancelled various meetings to secure social distancing and minimize potential damage. Furthermore, many activities that were undertaken through personal contact are being replaced by online engagement in organizations . Since it is possible that disasters such as the COVID-19 outbreak can re-occur, ICT skills for online work will become even more important, and middle-aged employees will need to raise their ICT capabilities beyond current levels; Continuous use of ICT technology can prevent the PSTRE from declining with age. Otherwise, they will face difficulties not only in their current jobs but also in re-employment.

Round 2

Reviewer 1 Report

Dear authors,

Thanks for your reply. Congratulations on your work.

Best regards

This manuscript is a resubmission of an earlier submission. The following is a list of the peer review reports and author responses from that submission.